# Self-Compassion and Well-Being during the COVID-19 Pandemic: A Study of Greek College Students

**DOI:** 10.3390/ijerph20064890

**Published:** 2023-03-10

**Authors:** Eirini Karakasidou, Georgia Raftopoulou, Anna Papadimitriou, Anastassios Stalikas

**Affiliations:** Department of Psychology, Panteion University of Social and Political Sciences, 176 71 Athens, Greece

**Keywords:** self-compassion, COVID-19, students, mental health, subjective happiness, life satisfaction

## Abstract

The present study examined the relationship between self-compassion (SC) and emotional well-being in college students during the COVID-19 pandemic. The theoretical framework for the study was that SC, defined as an understanding and caring response to one’s suffering and limitations, may serve as a protective factor against negative mental health outcomes. A sample of college students (N = 101) completed self-report measures of SC, depression, anxiety, stress, life satisfaction and subjective happiness. Data were analysed using regression analysis to examine the prediction of emotional well-being variables by SC and moderation analysis to examine the moderating effect of SC on the relationships between emotional well-being variables. The study’s results confirmed the hypothesis that SC would predict emotional well-being. SC significantly predicted all variables examined, including depression, anxiety, stress, life satisfaction (LS) and subjective happiness (SH). However, SC did not moderate the relationships between these variables. Isolation significantly moderated the relationship between SH and depression among college students. These findings support the idea that SC may serve as a protective factor against negative mental health outcomes and suggest that interventions aimed at increasing SC may improve mental health and overall well-being in college students during the COVID-19 pandemic. Further research is needed to understand these relationships’ mechanisms and the factors that may influence them.

## 1. Introduction

The COVID-19 pandemic has significantly impacted humanity globally, causing restrictions such as physical isolation and limitations on social contact [1]. This has resulted in changes to work and education, with students particularly affected by the shift to distance learning [2]. This has posed technological and financial difficulties and reduced university socialisation opportunities [3]. The restriction on activities and social isolation due to COVID-19 resulted in increased loneliness [4], anxiety, depression and stress among students [5], who also faced financial difficulties as they could not find jobs to meet their expenses [6]. Initially, during the pandemic, the literature focused more on the negative emotions experienced by students during quarantine, such as increased anxiety, stress and depression. However, researchers have recently started focusing more on positive emotions, recognising their role in protecting students’ mental health and moderating the impact of negative emotions. Given the widespread impact of the pandemic on mental health, it is increasingly important to understand the factors that may protect individuals against adverse outcomes. Therefore, the current survey aims to study the role of Self-Compassion (SC) in university students during COVID-19 and its relationship with anxiety, stress, depression, Life Satisfaction (LS) and Subjective Happiness (SH). To our knowledge, this is the first study to specifically examine the relationship between SC and emotional well-being in this population.

The mental health effects of quarantine on university students have been widely studied, and research has shown that students globally have experienced high levels of anxiety, stress and depression. Surveys from different countries worldwide report high anxiety, stress and depression among university students during the pandemic [7,8]. The research conducted on Greek university students shows that they experienced high levels of stress [9,10], anxiety [10] and depression [9,10] during the COVID-19 pandemic, similarly to students globally. Previous research on Greek students has found similar results, and it is a global phenomenon. Although most students had negative emotions, a proportion reported lower levels of anxiety, stress and depression. This raises questions about the skills and reasons that helped some students better manage the challenges of COVID-19 and maintain better mental health. Thus, the present study sheds light on positive components such as LS, SC and SH in students’ lives during COVID-19 in Greece. 

SC is a positive attitude towards oneself during difficult times, characterised by kindness, commonality of experiences and awareness of thoughts and feelings without criticism, rumination or avoidance [11]. The role of SC in managing difficulties during the COVID-19 pandemic has been studied in the general population. During COVID-19, studies have shown that SC predicts lower levels of anxiety, stress and depression and higher levels of well-being [12]. Previous research has also shown the positive impact of SC on mental health and LS [13]. People with higher SC experienced more hope and happiness [12]. SC also mediates the relationship between fear of COVID-19 and death anxiety [14] and anxiety about COVID-19 and body image disturbance [15]. However, it is essential to note that limited research has been conducted on students. One research conducted on students showed that SC predicted lower anxiety, stress and depression and higher levels of well-being [8]. The study also showed that SC improved their mental health but did not significantly contribute to academic engagement. Given the unique stressors and challenges posed by the pandemic, it is crucial to understand the role of SC in promoting positive mental health outcomes in this context. 

Studies have shown the protective role of SC in university students during COVID-19. The research found that SC mediates the relationship between maladaptive perfectionism and life satisfaction [16]. SC also moderates the relationship between perceived COVID-19 health risk and depression and the relationship between perceived impact and anxiety [17]. SC was found to be a protective factor that reduces stress and enhances well-being during COVID-19 [18] and has a positive relationship with well-being in students who did not work due to pandemic restrictions [6]. However, in another study [19], SC did not act protectively in the relationship between distress and well-being. Especially in women, it has been observed that body surveillance is negatively related to happiness in women with low SC [20]. 

During COVID-19, LS acted as a protective factor. According to [21], LS occurs when one’s life conditions match ideal and perfect conditions. However, COVID-19 brought restrictions, physical isolation, uncertainty and conditions that were far from ideal, leading to lower life satisfaction levels. A global survey with 35 research organisations found that 17% of participants reported feeling utterly dissatisfied with their lives, and 16% reported decreased LS [22]. Studies on the effect of COVID-19 on LS in university students have produced mixed results. Students diagnosed with COVID-19 or who had someone from their social environment with the virus had lower levels of LS [23]. However, students with higher self-efficacy and organisational identification felt more satisfied with life [24]. Students with supportive networks also had greater LS [25]. The relationship between LS, SH and SC during COVID-19 is limited. A positive relationship between SC and LS has been observed, with individuals who experience more SC feeling more satisfied with their lives [13]. This was also supported by a study on students, which showed that positive dimensions of SC (self-kindness, common humanity, mindfulness) were positively related to LS [26]. Meanwhile, negative dimensions such as self-criticism were negatively related to LS, while there was no statistical significance with the dimensions of isolation and over-identification [26].

SH is a dimension of well-being that has not been widely studied during COVID-19. According to [27], SH involves the achievement of goals (contentment) and the experience of pleasant moments (hedonism). The few studies on SH during COVID-19 had more promising results. The levels of SH among students were influenced by their perceptions of the educational environment [28], and those with social networks reported greater SH [25]. Students during COVID-19 experienced greater SH [29]. The relationship between SC, LS and SH among university students during COVID-19 is limited. Studies show a positive relationship between SC and LS, where individuals who experience more SC feel more satisfied with their lives [13,26]. Research on the relationship between SC and SH in university students during COVID-19 is limited. However, previous studies on young people and the general population suggest a positive relationship between SC and SH and a negative relationship between negative dimensions of SC and happiness [12,30]. Research on nursing students also shows that individuals who experience SC can better manage difficult situations and experience less anxiety, stress and depression [31]. According to the literature, limited studies have shown the relationship between anxiety, stress, depression and SH during COVID-19. One study on students in Brazil showed a negative relationship between anxiety, stress, depression and LS [7]. However, the relationship between these variables and SH during COVID-19 has not been studied. The only study that investigated this relationship was conducted before COVID-19 and was specific to pharmacy students, showing that stress, anxiety, and depression were negatively related to SH [32].

The literature has emphasised the protective role of SC during the COVID-19 pandemic. Studies have shown that SC moderates the negative effect of body surveillance on happiness and depression, especially among women with low SC [20]. It has also been a protective factor in reducing stress and enhancing well-being during the pandemic [18]. A negative relationship between SC and distress and a positive relationship with well-being has been confirmed in university students [19]. The relationship between SC and life LS has been studied in university students [13,16,26]. The relationship between SC and anxiety, depression and stress has been shown to be protective during the COVID-19 pandemic [17,18]. Although research on the relationship between SC and SH has been conducted before COVID-19 in students [32] and young people [30] as well as the general population [12], the relationship between SC and SH during COVID-19 in university students has not yet been studied. This gap in research is the focus of the present study. The relationship between anxiety, stress and depression and SH during the pandemic has not been studied. Still, it has been shown to be negatively related to SH in previous studies [32] and to subjective well-being [7] during the COVID-19 pandemic. 

Despite the growing body of literature on the relationship between SC and emotional well-being, there remains a gap in knowledge regarding the role of SC in promoting positive mental health outcomes in college students during the COVID-19 pandemic. Further research is needed to understand the specific mechanisms underlying these relationships and the factors that may influence them. The present study aims to fill this gap in knowledge by examining the relationship between SC and emotional well-being in a sample of college students during the COVID-19 pandemic. By examining the moderating effect of SC on the relationships between emotional well-being variables, this study will contribute to a better understanding of the role of SC in promoting positive mental health outcomes in this population. Therefore, the current survey aims to study the role of SC in university students during COVID-19 and its relationship with anxiety, stress, depression, LS and SH. The reasoning of the research is to shed more emphasis on the positive components of students during the COVID-19 period and highlight the protective role that can exist when we show compassion to ourselves. Additionally, from the literature review, there appeared to be fewer data regarding the relationship between SC with LS and SH during the period of COVID-19 and especially in university students, as most research focused on the relationship between SC with well-being. In addition, the protective role of SC in the relationship between anxiety, stress and depression with LS and SH has not been studied in university students during COVID-19, so it is a key motivation of the present survey.

The research hypotheses are:

(1) SC will predict anxiety, stress and depression during COVID-19 in university students.

(2) SC will predict LS and SH during the COVID-19 period in university students.

The research question is:

(1) Does SC moderates the relationship between anxiety, stress and depression with LS and SH during COVID-19 in university students?

## 2. Method

### 2.1. Participants

The research sample consists of 101 students, of which 81 are women, and 20 are men. The average age of the participants is 23.54 years (*SD* = 3.84). Participants were derived from public and private universities in Greece. Regarding the educational level of the sample, 21 people are postgraduate students (20.8%), and 80 are undergraduate students (79.2%). The current study compromises the ethics of psychology ethically. Participants provided volitional consent, and the recruitment method was snowball sampling. Private data were anonymous except for those required for the study, as no identifying information was collected. Participants were required to fill in the Self-Compassion Scale, DASS 21, Subjective Happiness Scale and Life Satisfaction Scale distributed in Google form through e-mails and Facebook. Their demographics are reported in Table 1. 

### 2.2. Materials

For the data generation, self-report questionnaires were administered, which recorded the following variables: demographic characteristics, Self–Compassion Scale, Subjective Happiness Scale, DASS 21 and Life Satisfaction Questionnaire. All questionnaires have good psychometric properties. Informed consent was provided. The demographic questionnaire includes age, sex, personal status, education level and employment status. Variables reflecting SC, anxiety, stress, depression, SH and LS were measured.

Self-Compassion Scale. SC was assessed by Self-Compassion Scale (SCS) [11]. The scale was translated into Greek [33]. The SCS assesses six aspects of SC: Self-Kindness, Self-Judgment, Common-Humanity, Isolation, Mindfulness and Over-identification. Each item was rated on a 5-point response scale ranging from 1 (Almost Never) to 5 (Almost Always). Mean scores are then averaged (after reverse-coding negative items) to create an overall SC score ranging from 26 to 130. Higher scores correspond to higher levels of SC. The standardisation of the Greek version of SCS showed satisfactory reliability and validity, and the factorial structure of the scale was found to match the ones from previous studies of other countries [33]. 

DASS-21. Depression, anxiety and stress symptoms were assessed with the short version of the DASS 21 (DASS 21, 21 items) [34]. In this study, Cronbach alphas were measured a = 0.94. The scale consists of 21 statements on a four-point Likert scale (0: Did not apply to me at all; 1,2,3: Applied to me very much or most of the time), referring to the previous week. 

SHS. The Subjective Happiness Scale documents a subjective perception of happiness (SHS, four items) [35], and it has been standardised in Greece [36]. In this study, the Cronbach alpha was measured at a = 0.88. Its four items are rated on a 7-point Likert scale. The lowest scores indicate a not very happy, and the highest scores indicate a very happy person. 

Life Satisfaction. Life Satisfaction Questionnaire measures an individual’s level of satisfaction, using a 7-point Likert scale ranging from “Strongly Disagree” to “Strongly Agree” (SWLS, five items) [37]. The Cronbach alpha of this study was a = 0.84. It has been standardised in the Greek sample population [38].

### 2.3. Procedure and Statistical Analyses

The Institutional Review Board approved this study in March 2021. The participants were recruited on the internet using electronic tools. All individuals are voluntary participants. Informed consent was provided. Participants completed a demographic questionnaire, SCS, DASS 21, Subjective Happiness Scale and LS. Individuals could withdraw at any time. Anonymity was ensured. 

This study used a non-experimental, cross-sectional design. An a priori power analysis was conducted using G*Power version 3.1.9.7 [39] to determine the minimum sample size required for testing the study hypothesis. The results indicated the required sample size to achieve 95% power for detecting a medium effect, at a significance criterion of α = 0.05, was n = 107 for multiple regression analysis. The obtained sample size of n = 104 is adequate to test the study hypothesis. Data processing was performed using the statistical package IBM SPSS Statistics (Version 26). We calculated the means, standard deviations, and Cronbach’s alpha internal reliability indices for all subscales administered. Then, multiple regression analyses were performed to examine SC’s contribution to positive and negative well-being variables. Finally, moderation analysis was conducted using the PROCESS tool between SC (as a total score) and positive and negative well-being factors in all possible combinations. 

## 3. Results

### 3.1. Descriptive Statistics

#### 3.1.1. Descriptive Statistics and Internal Reliability of Psychometric Tools

The results in Table 2 show the means, standard deviations and Cronbach’s alpha internal reliability indices for all subscales administered. All subscales had adequate to high reliability. This indicates that the sample group could be representative of the general population, as the levels of the measured variables do not deviate significantly from the norm. The data suggest that the participants in the study had low levels of psychopathology, moderate to high levels of LS and SH, and moderate levels of SC.

#### 3.1.2. Testing Research Questions and Hypotheses

(1) Predicting emotional well-being

In order to answer the first research question, regression analyses were performed with SC (as a total score to avoid multicollinearity) as a predictor variable of emotional well-being factors. The resulting statistically significant models are presented below.

Table 3 shows the prediction of SC on depression for college students. The value *R*^2^ = 0.28 revealed that the predictor variable explained 28% of the variance of the outcome predictor variable with *F* (1, 99) = 39.36, *p* < 0.001. The findings show that SC (β = −0.53, *p* < 0.001) negatively predicts college student depression.

Table 4 shows the prediction of SC on anxiety for college students. The value *R*^2^ = 0.16 revealed that the predictor variable explained 16% of the variance of the outcome predictor variable with *F* (1, 99) = 18.49, *p* < 0.001. The findings show that SC (β = −0.40, *p* < 0.001) predicts college student anxiety negatively.

Table 5 shows the prediction of SC on stress for college students. The value *R*^2^ = 0.17 revealed that the predictor variable explained 17% of the variance of the outcome predictor variable with *F* (1, 99) = 20.17, *p* < 0.001. The findings show that SC (β = −0.41, *p* < 0.001) predicts college student stress negatively.

Table 6 shows the prediction of SC on LS for college students. The value *R*^2^ = 0.20 revealed that the predictor variable explained 20% of the variance of the outcome predictor variable with *F* (1, 99) = 24.42, *p* < 0.001. The findings show that SC (β = 0.45, *p* < 0.001) positively predicts college student LS.

Table 7 shows the prediction of SC on SH for college students. The value *R*^2^ = 0.42 revealed that the predictor variable explained 42% of the variance of the outcome predictor variable with *F* (1, 99) = 71.59, *p* < 0.001. The findings show that SC (β = 0.65, *p* < 0.001) predicts college students’ SH positively.

#### 3.1.3. SC as a Moderating Variable in the Relationship between Body Image and Emotional Well-Being

To investigate the research question, moderation analyses were performed between SC (as a total score), positive well-being factors and negative well-being factors in all their possible combinations, using the PROCESS tool. The models that emerged are presented below Table 8. 

The interaction of SC as a total score and LS does not significantly predict depression, *b*= −0.01 CI [−0.02, 0.01], *t* = −0.87, *p* = 0.06, showing the relationship between LS and depression is not moderated by college student SC levels. The interaction of SC as a total score and SH does not significantly predict depression, *b*= −0.003 CI [−0.006, 0.012], *t* = 0.673, *p* = 0.50, showing the relationship between SH and depression is not moderated by college student SC levels. Similarly, all the analyses did not produce statistically significant results. 

In order to investigate whether specific factors of SC moderated the relationship between satisfaction with appearance and body image, regression analysis was conducted with each factor of SC individually. Only the statistically significant results are reported below.

Further analyses showed that the interaction between the factor of isolation and the factor of SH significantly predicts depression, *b* = 0.04, CI [−0.01,0.08], *t* = 1.71, *p* < 0.05, indicating that college student isolation levels moderate the relationship between SH and depression.

As found by the subsequent simple slopes analysis performed, when isolation levels are low, there is a significant negative relationship between SH and depression, *b* = −0.57, 95%, CI [−0.85, −0.30], *t* = −4.15, *p* = 0.001. At the mean of isolation levels, there is a significant negative relationship between SH and depression, *b* = −0.44, 95%, CI [−0.67, −0.21], *t* = −3.84, *p* = 0.002, which becomes non-significant at high levels of isolation, *b* = −0.31, 95%, CI [−0.59, −0.04], *t* = −2.24, *p* = 0.27.

From the above findings, it appears that the negative relationship between SH and depression is observed in individuals with low or moderate levels of isolation. Figure 1 presents the conceptual moderating model of the relationship between SH and depression, with college student isolation as a moderating variable.

## 4. Discussion

The present study examined the relationship between SC and emotional well-being among college students during the COVID-19 pandemic. The research aimed to highlight the positive aspects of students’ experiences during the pandemic and investigate the potential protective role of SC. The study’s findings support the growing body of literature indicating the protective role of SC on mental health outcomes [12,40]. Specifically, higher levels of SC predicted lower levels of depression, anxiety and stress, and higher levels of LS and SH. This confirms the hypothesis that SC may serve as a protective factor against adverse mental health outcomes during the pandemic. However, the current study also sheds light on a gap in the literature by demonstrating that SC did not moderate the relationships between emotional well-being variables, despite previous studies indicating a potential moderation effect [4,13,14]. Instead, isolation was found to moderate the relationship between SH and depression. This finding challenges previous assumptions and suggests that the relationship between SC and emotional well-being may be more complex than previously thought. It also highlights the importance of considering college students’ context and individual experiences during the pandemic and the role that different factors may play in their mental health and well-being.

Synthesising findings from the literature, we hypothesised that college students’ emotional well-being would be predicted by their levels of SC. This hypothesis was confirmed regarding predicting all variables, namely, depression, anxiety, stress and LS and SH. In particular, SC was found to negatively predict depression, anxiety and stress, while it positively predicted LS and SH. This finding is supported by a growing body of research on the positive effects of SC on mental health, including research related explicitly to the COVID-19 pandemic. For example, a study [40] found that SC was significantly associated with lower levels of depression, anxiety and stress, and higher levels of well-being, among college students during the pandemic. Another study [12] found that SC was significantly related to better mental health outcomes, including lower levels of depression and anxiety and higher levels of well-being, among a sample of college students in China during the pandemic.

To further explore the potential protective role of SC on the emotional well-being of college students, the researchers investigated the moderating role of SC with the abovementioned variables. It was found that SC did not moderate the relationship between LS, SH, depression, anxiety or stress. According to the literature review, no other research has been conducted on the regulatory role of SC among these variables. However, this finding contrasts the results of previous research, which has consistently demonstrated that SC is related to better mental health outcomes, including lower levels of depression, anxiety and stress, and higher levels of LS and SH [11]. It is possible that the current study did not find a moderation effect because the sample size may not have been sufficient to detect small but meaningful differences or because the measures used in the study were not sensitive enough to capture the moderating effects of SC.

In addition to its novel findings, this study builds on the strengths of previous research by exploring the regulatory role of SC. This area has received limited attention in the literature. Furthermore, this study provides a unique perspective on the complex relationship between self-compassion, emotional well-being, and the COVID-19 pandemic, offering insights into the potential protective role of SC during times of stress and uncertainty. It is also worth considering that SC may not always operate as a moderating factor in the relationships between mental health and well-being variables. For example, SC may be more strongly related to some mental health outcomes (e.g., depression and anxiety) than others (e.g., LS), or it may have different effects in different contexts or populations. Further research is needed to explore how SC may or may not moderate the relationships between mental health and well-being variables. Further research is needed to understand these relationships’ mechanisms and the factors that may influence them.

One of the key findings of our research was that isolation significantly moderated the relationship between SH and depression in college students during COVID-19. Specifically, it was found that college students who reported lower levels of isolation during the pandemic had higher levels of SH and lower levels of depression. To our knowledge, no other research has investigated the moderating role of isolation between SH and depression in college students. Regarding the factors of SC, it was found that the isolation factor moderates the relationship between SH and depression, which has not yet been investigated. For college students, it was found that for those who presented low or moderate levels of isolation, SH could negatively predict depression. On the contrary, no statistically significant result was reported for college students with high isolation levels.

Previous research has established that physical isolation can negatively impact mental health, including an increased risk of depression [41]. The current study supports these findings, as we observed a negative correlation between isolation and SH and a positive correlation between isolation and depression. These findings are consistent with previous research on the adverse effects of physical isolation on mental health. A review of the literature on physical isolation and depression found that isolated individuals are at an increased risk for developing depression and that social support can serve as a protective factor against the onset of depression [4]. Similarly, research has shown that social connections and positive relationships are important determinants of SH [4,17]. However, our results also suggest that SC may serve as a protective factor against the negative effects of isolation on mental health. We found that SC was significantly correlated with both SH and lower levels of depression. Moreover, when SC was included in the model as a mediator, it significantly reduced the relationship between isolation and SH and depression.

## 5. Limitations and Future Directions

The results of this study suggest that SC predicts depression, anxiety, stress, LS and SH. In addition, isolation plays a significant role in mediating the relationship between SH and depression in college students during the COVID-19 pandemic. Our findings align with previous research demonstrating the beneficial effects of SC on mental health outcomes and highlight the importance of addressing physical isolation and promoting SC as strategies for supporting mental health and well-being in this population. However, it is essential to consider the limitations of any research study in order to interpret and understand the results accurately. A potential limitation of the current study includes the sample size. The sample size for this study may not have been large enough to fully capture the diversity of experiences and perspectives within the population of college students during the COVID-19 pandemic. A larger sample size would allow for more robust analysis and may reveal other patterns and relationships. We did not use sex as a variable in the statistical analysis because of the sex imbalance in our sample (19.8% male). Females with SC have been found to adopt more positive coping strategies leading to higher LS [13].

Moreover, the measures used in this study rely on self-report, which can be subject to biases such as social desirability or memory errors. Multiple methods (e.g., self-report, observation and physiological measures) could provide a more comprehensive understanding of the studied variables. In addition, the study’s cross-sectional design limits the ability to make conclusions about causality or the direction of the relationships between variables. Finally, the external validity of the results should be considered, as it is possible that the findings may not be generalisable to other populations or contexts. Finally, the external validity of the results should be considered, as it is possible that the findings may not be generalisable to other populations or contexts. Overall, these limitations should be taken into consideration when interpreting and applying the results of the current study. 

Future research should aim to replicate and expand upon these findings and explore potential interventions for promoting SC and reducing physical isolation in college students during the COVID-19 pandemic. Future research should address the current study’s limitations and build upon its findings. A larger sample size would provide more robust results and allow for the analysis of different subgroups within the population of college students during the COVID-19 pandemic. Longitudinal designs would enable examining the temporal relationships between variables and the impact of SC and isolation over time. Using multiple methods (e.g., self-report, observation, physiological measures) would provide a more comprehensive understanding of the studied variables and reduce the potential biases associated with self-report measures. Future research should aim to examine the efficacy of potential interventions for promoting SC and reducing physical isolation in college students during the COVID-19 pandemic. This could involve developing and testing interventions to increase SC, reduce isolation, and explore the potential moderators of these interventions (e.g., age, sex and culture). This study was conducted among Greek college students. Cross-cultural comparisons would provide valuable insight into the generalisability of the findings and the potential cultural and contextual factors affecting the relationship between SC, isolation and emotional well-being. 

## 6. Contribution to Scientific Knowledge and Therapeutic Practice

The results of this study are significant because they provide novel insights into the relationship between SC and emotional well-being in a time of heightened stress and uncertainty, such as during the COVID-19 pandemic. College students, already facing the challenges of a rapidly changing world, have been disproportionately impacted by the pandemic. The findings contribute to a growing body of literature on SC and mental health and have implications for developing interventions to improve mental health and overall well-being in college students during the COVID-19 pandemic. In particular, the study highlights the potential role of increasing SC as a protective factor against adverse mental health outcomes and underscores the importance of considering individual experiences and contextual factors in developing effective interventions. Understanding the role of SC in promoting emotional well-being in this population is crucial for developing effective interventions to improve mental health.

The findings of the current research align with prior research demonstrating the beneficial effects of SC on mental health outcomes (Neff, 2003). They also highlight the importance of addressing physical isolation and promoting SC as strategies for supporting mental health and well-being in college students during the COVID-19 pandemic to help students manage feelings of isolation and loneliness. The mediating role of isolation in the relationship between the COVID-19 pandemic and SH and depression suggests that interventions aimed at reducing isolation and promoting social connection may effectively mitigate the pandemic’s adverse mental health effects on college students. For example, virtual social support groups, online social activities, and phone or video call check-ins with friends and family may help to alleviate feelings of isolation and promote SH and well-being. In conclusion, our research findings highlight the importance of addressing SC as a potentially protective factor against the adverse mental health outcomes experienced by college students during the COVID-19 pandemic. Future research could be guided towards developing and evaluating interventions for promoting SC among college students. Furthermore, future research could be guided towards reducing isolation and promoting social connection. 

## 7. Conclusions

In conclusion, the present study examined the relationship between SC and emotional well-being in college students during the COVID-19 pandemic. The results confirmed our hypothesis that SC would predict emotional well-being, as SC significantly predicted all variables examined, including depression, anxiety, stress, LS and SH. This finding is supported by a growing body of research on the positive effects of SC on mental health, including research related explicitly to the COVID-19 pandemic. One key finding from our study is that SC does not moderate the relationship between LS, SH, depression, anxiety or stress. This means that SC does not significantly alter the strength or direction of the relationships between these variables. Another key finding from our study is that isolation moderates the relationship between SH and depression.

All in all, his study highlights the importance of further research into the role of SC in promoting emotional well-being, particularly during times of stress and uncertainty. By challenging previous assumptions and offering novel insights, this study contributes to the growing body of literature on the positive impact of SC on mental health and well-being. Overall, the results of this study provide evidence for the beneficial role of SC in mitigating the negative impacts of isolation on SH and depression in college students during the COVID-19 pandemic. Further research is needed to replicate and expand upon these findings and explore potential interventions for promoting SC and reducing the effects of physical isolation in this population.

## Figures and Tables

**Figure 1 ijerph-20-04890-f001:**
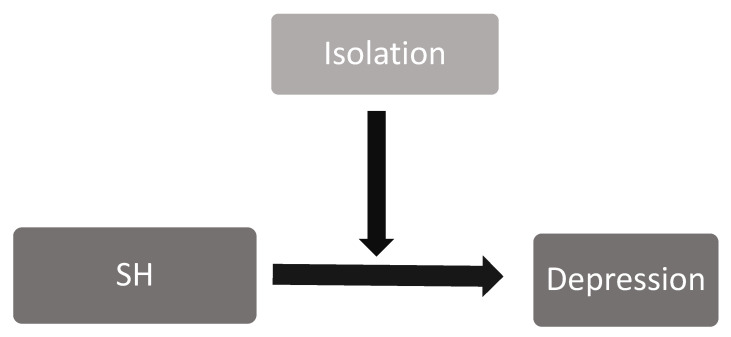
Diagram of the Moderating Conceptual Model of the Relationship between SH and Depression with college student isolation as moderator.

**Table 1 ijerph-20-04890-t001:** Demographic characteristics of the participants (N = 101).

Demographic Variables	University Students (N = 101)
Age (Mean, SD)	23.54 (3.84)
Sex	
Male	20 (19.8%)
Female	81 (80.2%)
Marital status	
Single	85 (84.2%)
Married	16 (15.8%)
Education	
College student	80 (79.2%)
Postgraduate student	21 (20.8)
COVID diagnosis	
Diagnosed	10 (9.9%)
Not diagnosed	91 (91.0%)
Family member’s diagnosis	
Diagnosed	90 (89.1%)
Not diagnosed	11 (10.9%)

**Table 2 ijerph-20-04890-t002:** Reliability indices, means and standard deviations of all scales and subscales.

Subscales	Cronbachs’α	M	SD
Self-kindness	0.79	16.11	3.67
Self-judgement	0.85	14.15	4.51
Common humanity	0.76	13.30	3.24
Isolation	0.70	13.56	2.78
Mindfulness	0.71	13.56	2.78
Over-identification	0.70	12.60	3.25
SC	0.92	83.98	16.53
LS	0.83	22.54	5.72
SH	0.86	18.49	5.04
Depression	0.89	7.18	5.32
Anxiety	0.86	4.37	4.30
Stress	0.88	8.49	4.86

**Table 3 ijerph-20-04890-t003:** Simple Linear Regression for depression (Ν = 101).

Variable	*B*	*SE*	*t*	*p*	95% CI
Constant	21.38	2.31	9.26	0.00	[16.46, 25.68]
SC	−0.18	0.03	−6.27	0.00	[−0.23, −0.12]

**Table 4 ijerph-20-04890-t004:** Simple Linear Regression for anxiety (Ν = 101).

Variable	*B*	*SE*	*t*	*p*	95% CI
Constant	12.91	2.03	6.32	0.00	[8.73, 16.72]
SC	−0.11	0.03	−4.28	0.00	[−0.16, −0.06]

**Table 5 ijerph-20-04890-t005:** Simple Linear Regression for stress (Ν = 101).

Variable	*B*	*SE*	*t*	*p*	95% CI
Constant	18.51	2.28	7.96	0.00	[13.69, 22.77]
SC	−0.13	0.03	−4.40	0.00	[−0.18, −0.07]

**Table 6 ijerph-20-04890-t006:** Simple Linear Regression for LS (Ν = 101).

Variable	*B*	*SE*	*t*	*p*	95% CI
Constant	9.79	2.63	3.77	0.00	[4.62, 14.87]
SC	0.16	0.03	5.06	0.00	[0.10, 0.23]

**Table 7 ijerph-20-04890-t007:** Simple Linear Regression for SH (Ν = 101).

Variable	*B*	*SE*	*t*	*p*	95% CI
Constant	2.12	1.97	1.32	0.00	[−1.31, 6.5]
SC	0.21	0.02	8.23	0.00	[0.15, 0.25]

**Table 8 ijerph-20-04890-t008:** Linear Model of Predictors of Depression: Isolation and SH.

Variable	*B*	*SE*	*t*	*p*	95% CI
SH	−0.44	0.12	−3.84	0.000	[−0.67, −0.21]
Isolation	−0.41	0.15	−2.76	0.006	[−0.01, 0.08]
SH × Isolation	−0.04	0.02	1.71	0.008	[−0.01, 0.08]

## Data Availability

The study’s data are available on request from the corresponding author. The data are not publicly available due to restrictions on privacy.

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
