# Peer review of "Self-Compassion and Well-Being during the COVID-19 Pandemic: A Study of Greek College Students"

_ijerph, 2023, doi:10.3390/ijerph20064890_

Round 1
Reviewer 1 Report
Dear Authors,
Your manuscript about Self-compassion and well-being during the pandemic of COVID – 19 was about the effect of self-compassion on
depression, anxiety, stress, life satisfaction, and subjective happiness, and the moderator role of Isolation needs to be improved.
The introduction needs to be improved. The literature review does not support your main question as well.
The literature review needs to reorganize. Using abbreviations has been recommended to summarize it.
The sample size is not adequate for this type of research.
Gender imbalance in the sample (a convenient sample) is not acceptable.
The analysis ignored gender as a variable.
The manuscript needs literacy editing.
Author Response
Please find attached our response.

Author Response
Please find attached our response.

Reviewer 3 Report
Authors propose to analyse self-compassion and well being during COVID-19 pandemic. This is a study already conducted with other groups, but not with college students.
It is important to clarify that due to pandemic crisis one important measure to control the virus spread was physical isolation, not social isolation. Social isolation is not bearable for us, humans as social beings.
Results point that “self-compassion may serve as a protective factor against negative mental health” which follows the expected trend… It looks like there are no novelty with this study, little knowledge is added.
Author Response
Attached please find my response to your comments.

Round 2
Reviewer 1 Report
Dear authors,
The manuscript needs minor revision for language
Author Response
Thank you very much for your comments!
The article has been once again proofread using the professional version of Grammarly.
Reviewer 3 Report
While I acknowledge the effort of the authors in reviewing the manuscript, I stand by my opinion regarding the aspects pointed out as weaknesses. Therefore, I maintain the decision to reject the article for publication in ijerph.
Author Response
Thank you very much for your comments.
I have made the necessary changes, and I hope they meet your expectations.